# *In Vitro* Termiticidal Activity of Medicinal Plant Essential Oils Against *Microcerotermes crassus*

**DOI:** 10.3390/insects16121261

**Published:** 2025-12-11

**Authors:** Chaiamon Chantarapitak, Jarongsak Pumnuan, Chaiwat Chanpitak, Somsak Kramchote

**Affiliations:** 1School of Agricultural Technology, King Mongkut’s Institute of Technology Ladkrabang, Bangkok 10520, Thailand; 2Union of Unicor Group Co., Ltd., 88, Nawamintr 24 Rd., Bueng Kum District, Bangkok 10240, Thailand

**Keywords:** termites, clove, cinnamon, star anise, toxicity, repellency

## Abstract

Termites are among the most destructive pests affecting both households and agriculture, leading to substantial economic losses worldwide. Although conventional chemical termiticides are effective, their extensive use raises concerns regarding environment condition and human health risks. This study investigated the termiticidal and repellent activities of essential oils extracted from clove (*Syzygium aromaticum*), cinnamon (*Cinnamomum zeylanicum*), and star anise (*Illicium verum*) against *Microcerotermes crassus* under controlledlaboratory conditions. The findings demonstrated that clove and cinnamon oils, both rich in eugenol, exhibited strong acute toxicity and repellency, resulting in complete termite mortality at 250 µL/L. In contrast, star anise oil and its principal compound, anethole, showed lower repellency but delayed yet sustained lethal effects. These characteristics suggest that eugenol-based oils are suitable candidates for direct-contact spray formulations, whereas anethole-based oils may be more appropriate for use in slow-acting bait systems. Overall, this study highlights the potential of plant-derived essential oils as eco-friendly alternatives for termite management, contributing to the development of safer and more sustainable pest control strategies.

## 1. Introduction

Termites are economically significant pests that cause severe damage to agriculture, forestry, and wooden structures worldwide. More than 3000 species have been reported globally, of which at least 183 are recognized as pests of economic importance, particularly in tropical and subtropical regions [1,2,3]. *Microcerotermes* species are widespread wood-feeding termites commonly found in both natural and human-disturbed habitats through Thailand. They construct small subterranean or arboreal carton nests and forage through extensive mud-tube networks, enabling persistent attacks on structural timber, orchard trees, and crops. A nationwide survey identified *Microcerotermes crassus* as one of the most frequently encountered species, reflecting its ecological dominance and its importance as a structural and agricultural pest [4]. Their cryptic nesting habits and dispersed foraging galleries make infestations difficult to detect at early stages, often resulting in progressive and long-term damage before control measures can be implemented. Their organized caste systems and capacity to mobilize large workforces enable efficient attacks on plant roots, crop fields, and structural timber, while rapid colony expansion and permanent mounds make infestations persistent and difficult to control [5,6,7].

Traditional termite control in Thailand continues to rely on synthetic insecticides, particularly fipronil and cypermethrin, with their effectiveness supported by multiple field evaluations in the region [8,9,10]. However, these chemicals raise environmental concerns substantiated by quantitative data: fipronil shows soil half-lives of ~125–128 days and can exceed 200 days [11], whereas cypermethrin typically has soil DT_50_ ≈ 20–30 days but field DT_50_ spanning ~14–199 days depending on soil [12,13]. In aquatic systems, it photodegrades relatively quickly (water DT_50_ ≈ 3–8 d; water–sediment ≈ 17 d), yet residues can accumulate in the sea-surface microlayer and have been detected at 0.01–0.3 µg L^−1^—levels affecting sensitive biota; both compounds are highly toxic to benthic invertebrates and soil organisms, and extremely toxic to pollinators that are beneficial to nature [11,14,15,16,17]. Field evidence also shows rapid fipronil resistance in German cockroaches (3–10× LC_99_ tolerance shortly after bait introduction; ~14–15× cross-resistance; ≈36× increases in LD_50_) [18,19,20]. These data show the need for safer, more sustainable options.

Plant extracts are gaining attention as safer alternatives because they degrade quickly in the environment, are low-toxic to mammals at practical doses, and exhibit multi-component bioactivity [21,22,23]. However, their safety is dose-dependent, and several studies have demonstrated that eugenol can exert toxic effects at elevated concentrations; for example, impairing embryonic development in fish [24] and inducing acute or subacute toxicity in mammalian models when administered at high doses [25]. Many previous studies reported that cinnamon bark, clove, and other botanicals against stored-product pests [26,27,28]. In addition, the hexane crude extract from *Acorus calamus* rhizomes and its major compounds also exhibited significant termiticidal activity against *Coptotermes curvignathus*, as reported by Adfa et al. [29]. Similarly, essential oils (EOs) of *Illicium verum* and *Pimpinella anisum*, dominated by (E)-anethole, can suppress feeding and induce high mortality in species such as *Sitophilus zeamais*, *Callosobruchus maculatus*, and even the German cockroach under fumigation [30,31,32].

For termites, EO-based approaches have shown promising potential in laboratory study conditions, for example, fumigant toxicity of garlic and clove bud oils against *Reticulitermes speratus* and activities of conifer-derived EOs against *Coptotermes formosanus* [33,34]. Most previous studies focused only on short-term contact or fumigation tests, seldom targeted tropical taxa of high economic importance such as *Microcerotermes*, rarely linked detailed chemical profiles to measured bioefficacy, and infrequently benchmarked EOs against commercial termiticides under field-relevant conditions. Therefore, this study tests EOs from clove and cinnamon (with additional candidate plants) against *Microcerotermes* workers, using contact toxicity and repellency assays conducted in a closed system, GC–MS/MS profiling, and comparative performance against synthetic standards to generate evidence linking composition to practical efficacy for integration into Integrated Pest Management (IPM).

## 2. Materials and Methods

### 2.1. Essential Oils, Reference Compound, and Preparation

Essential oils (EOs) from 14 medicinal plants (Table 1) were from Thai-China Flavors and Fragrances Industry Co., Ltd. (Nonthaburi, Thailand), a facility operating under a Hazard Analysis and Critical Control Point (HACCP) program. Each EO was supplied in a sealed aluminum bottle; the botanical source, plant part, and supplier batch/lot number were recorded. Eugenol (≥99% purity), the main compound in clove and cinnamon oils, and anethole (≥99% purity), the key constitutes in star anise oil, were purchased from Sigma-Aldrich (St. Louis, MO, USA) and used as reference compounds.

### 2.2. Stock and Working Solutions

For aqueous dispersion, each essential oil and purified chemical standards (eugenol and anethole) were premixed with polysorbate-20 (Tween-20; Sigma-Aldrich) at a 1:2 (*v*/*v*) EO: Tween-20 ratio, then brought to volume with deionized water to obtain a 10% (*v*/*v*) stock. Fresh working solutions at the target assay concentrations were prepared by serial dilution in deionized water. A surfactant control (Tween-20 only at the highest final level used) and a water blank were included in each assay. All solutions were kept in amber glass vials with PTFE-lined caps, protected from light at 4 °C, and gently inverted immediately before use; dilutions were prepared on the day of the assay.

### 2.3. Termite Collection and Maintenance

Worker termites of *Microcerotermes crassus* were collected from natural mounds in Bangkapi District, Bangkok, Thailand. To protect the colony, only the outer parts of the mound were taken, leaving the queen inside the original mound. The collected nest material was placed in 20 L plastic boxes provisioned with moistened corrugated cardboard as food and shelter and transported to the laboratory at the School of Agricultural Technology, King Mongkut’s Institute of Technology Ladkrabang (KMITL), Thailand. Bioassays were conducted within 7 days of collection, and workers measuring 4.7 ± 0.4 mm in body length were selected. When additional termites were required, fresh material was collected from the same mound.

### 2.4. Chemical Characterization of Essential Oils

A preselected subset of EOs that exhibited high activity in preliminary screening assays (see Screening bioassays) was characterized by gas chromatography–mass spectrometry (GC–MS; Agilent 8890 GC coupled to a 5977-mass selective detector, Agilent Technologies, USA) fitted with an HP-5MS capillary column (30 m × 0.25 mm i.d. × 0.25 µm film). Samples were dissolved in HPLC-grade ethanol and 1.0 µL was injected in split mode (100:1). The injector was set to 250 °C; helium served as the carrier gas at 1.0 mL min^−1^ (constant flow). Electron-impact ionization operated at 70 eV with a scan range of *m*/*z* 50–600 and a 3 min solvent delay. The oven program was 40 °C (hold 3 min), ramp 10 °C min^−1^ to 150 °C (hold 5 min), then 15 °C min^−1^ to 260 °C (hold 5 min); the MS detector was maintained at 270 °C. Tentative identifications were assigned by comparison with the NIST20 mass spectral library, accepting matches ≥85%. Relative composition was reported as the percentage of the total ion chromatogram peak area (% area).

### 2.5. Contact Toxicity Bioassays

#### 2.5.1. Screening Bioassays

Contact-residue screening under sealed-dish conditions followed Pumnuan et al. [35] with minor adjustments. A Whatman™ No. 1 filter paper (9 cm diameter) was placed in a 9 cm glass Petri dish. Each dish received 1.0 mL of an EO working solution at 0.05, 0.1, and 1% (*v*/*v*) (i.e., 500, 1000, and 10,000 µL/L). The solution was spread evenly on the filter paper. Ten worker termites were introduced per dish immediately after treatment. Dishes were closed and maintained at room temperature. Percent mortality at 3 h was recorded; control dishes received 2% (*v*/*v*) Tween-20 in water only. For prioritization, EO performance at 3 h was classified as very low (<10%), low (10–25%), medium (26–50%), high (51–75%), or very high (>75%) mortality. Termite workers were classified as dead when they were unable to stand or walk after gentle probing with a fine hairbrush, even if slight antennal or leg movements remained, provided that this incapacity continued for at least 30 s. The top-performing EOs (2–3 candidates) were advanced to chemical characterization and to subsequent toxicity and repellency assays.

#### 2.5.2. Toxicity Assays (Contact-Residue Under Sealed-Dish Conditions, Follow-Up to Screening)

The top 2–3 EOs from the screening step were evaluated for dose–response toxicity using the same contact-residue setup described above. Working concentrations were 0.01–0.1% (*v*/*v*) (i.e., 100–1000 µL/L). Assays were conducted at room temperature, and percent mortality at 3, 6, 12, 18, and 24 h post-treatment was recorded. Control units received Tween-20 in water matched to the highest final level in the 0.1% EO treatment (0.2% *v*/*v*; EO: Tween-20 = 1:2). Eugenol and Anethol standards served as a reference compound, and fipronil (250 mg/L) and cypermethrin (2000 mg/L) at label-recommended rates. Data analysis: concentration–mortality and time–mortality data were used to calculate LC_50_/LC_90_ and LT_50_/LT_90_ by probit analysis. The experiment followed a completely randomized design (CRD) with three replicates per treatment.

### 2.6. Repellency Bioassays

EOs selected from the preceding contact-residue under sealed-dish conditions screen were evaluated for repellency in a CRD at 50, 100, and 500 µL/L with three replicates per treatment, using a modified half–filter paper residue test (after Ruddit et al. [36]). Whatman™ No. 1 filter papers (9 cm diameter) were halved; one half was treated with the EO solution and the other with a surfactant-matched control (Tween-20 in water at the same final Tween-20 level as the EO solution). Each half received 0.5 mL of its respective solution before reassembling in a 9 cm Petri dish. Ten worker termites were placed at the center; Dishes were sealed and kept at 25 ± 2 °C. Counts on the treated and control halves were recorded at 15, 30, 45 min; 1, 3, 6, 12, 18, and 24 h. Repellency was assessed exclusively from the spatial distribution of live termites on the treated and untreated halves. Because repellent observations were repeatedly recorded on the same group of 10 workers within each dish, the time-series measurements are not strictly independent. This limitation was considered when interpreting temporal patterns. Repellency (% Repelled = 100 × [number on the control half/10] per observation) was analyzed by fitting a binomial GLM (logit link) with treatment, concentration, time, and their interactions as fixed factors [37]. To mitigate complete separation at 0% or 100% repellency, a minor continuity correction (±0.1 individuals) was applied before model fitting [38]. Predicted values were back-transformed to the probability scale and reported as percentages, with 95% CIs based on model standard errors. Post hoc pairwise comparisons of estimated marginal means (EMMs) were performed using Tukey’s HSD adjustment. All analyses were performed in R 4.x [39] using the stats, emmeans, and DHARMa.

### 2.7. Data Analysis

Mortality screening at 3 h was categorized into five levels to select essential oils for further testing. Analyzed by ANOVA, followed by DMRT (α = 0.05), and probit analysis was applied to estimate the median lethal concentration (LC_50_), 90% lethal concentration (LC_90_), median lethal time (LT_50_), and 90% lethal time (LT_90_); only point estimates are reported because stable 95% CIs were not available. Repellency was assessed using a binomial GLM (logit) with likelihood-ratio χ^2^ tests and Holm-adjusted pairwise comparisons (continuity correction ±0.1 for 0/100% cases). For repellency, post hoc pairwise comparisons were performed on estimated marginal means (EMMs) using Tukey’s HSD adjustment to determine statistically homogeneous groups, which are presented as letter groupings above each treatment line (*p* < 0.05). All analyses were conducted in R 4.x (stats, emmeans, DHARMa), with significance set at α = 0.05.

## 3. Results

### 3.1. Screening and Chemical Composition of Essential Oils

In the first screening, 14 medicinal plant EOs were tested against *M. crassus* workers at 3 h. Several oils showed very strong activity, with cinnamon (*Cinnamomum zeylanicum*) and clove (*Syzygium aromaticum*) consistently caused >75% mortality at all concentrations (Table 1). Star anise (*Illicium verum*) also performed well, whereas anise (*Pimpinella anisum*) was weaker and less consistent. Based on these results, cinnamon, clove, and star anise oils were selected for further chemical characterization.

GC–MS/MS analysis revealed that clove oil was dominated by eugenol (66.69%), followed by caryophyllene (17.34%), α-bisabolene (6.22%), cadinene (2.74%), and several minor compounds (<5%). Cinnamon oil also had eugenol as the major component (54.99%), together with caryophyllene (7.22%), cinnamyl acetate (3.88%), humulene (2.06%), and benzyl benzoate (5.66%). In contrast, star anise oil was dominated by anethole (90.78%), with small amounts of linalool (1.39%), anisic aldehyde (1.81%), and α-copaene (0.81%) (Table 2). These results highlighted that clove and cinnamon oils are eugenol-rich, whereas star anise oil is dominated by anethole, providing distinct chemotypes for subsequent bio-efficacy testing.

### 3.2. Contact Toxicity and Lethal Time of Essential Oils

Following the screening, dose–response assays were conducted to determine lethal concentrations (LC_50_/LC_90_) and lethal times (LT_50_/LT_90_) for the selected EOs, compared with eugenol and anethole standards and commercial termiticides (Table 3 and Table 4). The clove oil was the most potent, achieving 100% mortality within 6 h at 250 µL/L, with LC_50_ values declining from 208.60 µL/L at 3 h to 70.60 µL/L at 24 h, and an LT_50_ of only 2.08 h. The cinnamon oil was slower, reaching 100% mortality at 250 µL/L after 24 h, with an LC_50_ of 130.50 µL/L and LT_50_ of 9.31 h. The eugenol standard showed a similar trend to cinnamon, with LC_50_ and LT_50_ values of 104.50 µL/L and 8.07 h, respectively. In addition, the star anise oil (anethole-rich) had moderate toxicity. At 250 µL/L it achieved only 43.30% mortality after 24 h, with LC_50_ values decreasing from 507.00 µL/L at 3 h to 332.80 µL/L at 24 h, and an LT_50_ of 11.98 h. The anethole standard performed comparably, with LC_50_ declining from 418.80 µL/L at 3 h to 216.60 µL/L at 24 h, and an LT_50_ of 13.14 h. These results indicated that although star anise oil and anethole exhibited measurable toxicity, their effects were weaker and slower than those of eugenol-rich oils. Whereas, the commercial insecticides, fipronil at 250 mg/L caused complete mortality within 12 h (LT_50_ = 3.35 h), while cypermethrin at 2000 mg/L achieved 100% mortality within the first 3 h of exposure.

Clove oil exhibited markedly greater toxicity against *M. crassus* than cinnamon oil, star anise oil, eugenol, and anethole. At 100 µL/L, clove oil caused more than 90% mortality within 24 h, a level comparable to that at 250 µL/L, showing that its activity was both strong and stable even at lower doses. In contrast, cinnamon oil, star anise oil, eugenol, and anethole required 250 µL/L to reach higher mortality, showing significantly weaker efficacy at 100 µL/L (*p* < 0.05). Moreover, clove oil exceeded 95% mortality at 250 µL/L within 6 h, with no substantial increase thereafter, demonstrating its rapid onset of action. These results highlight the superior potency and faster onset of action of clove oil compared to the slower, concentration-dependent responses of the other EOs and standards. For commercial insecticides, fipronil (250 mg/L) achieved 100% mortality within 12 h, while cypermethrin (2000 mg/L) caused complete mortality within 3 h, indicating stronger and faster effects than the EOs.

### 3.3. Repellent Activity of Essential Oils

The repellent activity of clove and cinnamon oils against *M. crassus* workers showed a clear dose- and time-dependent trend compared with the eugenol standard (Figure 1). At 50 µL/L, all three treatments produced moderate repellency (60–80%) during the first hour, with clove and cinnamon oils maintaining >70% up to 12 h, while eugenol declined below 60% at 24 h. At 100 µL/L, clove and cinnamon oils exceeded 80% repellency after 6 h and sustained their effects to 24 h, whereas eugenol dropped below 60%. At 500 µL/L, both clove and cinnamon oils exhibited the strongest activity, achieving >90% repellency within 3 h and maintaining it until 24 h, consistently surpassing eugenol. Post hoc comparisons (Tukey’s HSD, *p* < 0.05) confirmed that clove and cinnamon oils consistently formed the highest-repellency group at all concentrations, whereas star anise and anethole constituted the lowest group. Eugenol standard occupied an intermediate position. These statistical groupings are presented above each treatment line in Figure 1.

In contrast, star anise oil and the anethole standard exhibited noticeably weaker repellency. At 50 µL/L, both produced <60% repellency in the first hour, and their effects declined steadily afterward. At 100 µL/L, repellency fluctuated around 40–50% with no improvement over time. Even at 500 µL/L, star anise oil and anethole failed to exceed 60% repellency throughout the 24 h observation period. These findings indicate that clove and cinnamon oils are far more effective repellents than star anise oil and anethole.

## 4. Discussion

The contact-residue screening bioassays conducted under sealed-dish conditions revealed that clove (*S. aromaticum*) and cinnamon (*C. zeylanicum*) EOs were the most effective against *M. crassus* workers, consistently producing very high mortality (>75%) at all tested concentrations. Star anise (*Illicium verum*) also showed strong but less consistent effects, suggesting that its bio-efficacy may be more variable. The GC strongly supported these outcomes–MS/MS analyses, which showed that clove and cinnamon oils were rich in eugenol (66.69% and 54.99%, respectively). This is consistent with several previous studies of clove and cinnamon EOs [27,28,33,40]. In addition, star anise oil was dominated by anethole (90.78%), consistent with previous reports that identified anethole as the major constituent of both star anise and anise EOs [30,31,32].

The predominance of eugenol in clove and cinnamon oils provides a compelling mechanistic explanation for their superior toxicity. Eugenol is well known for its vigorous insecticidal activity. Huang et al. [40] showed that eugenol significantly disrupted feeding and caused high mortality in maize weevil (*Sitophilus zeamais*). Similarly, Plata-Rueda et al. [28] found that eugenol and related compounds in clove and cinnamon oils caused substantial toxicity and repellency against the granary weevil (*Sitophilus granaries*). Consistent with earlier reports, our results confirmed that eugenol-rich oils were more effective against termites compared with star anise and anise EOs, which mainly contain anethole. Chang and Ahn [30] and Wei et al. [31]. Although previous studies have reported strong insecticidal effects of star anise and anise, characterized by high levels of anethole, against stored-product beetles [31,32] and cockroaches [30], our findings suggest that these effects do not apply as strongly to termites, since *M. crassus* showed weaker and less consistent responses.

These differences highlight the importance of both target specificity and chemical composition in determining bioactivity. Anethole is known for its fumigant and anti-feeding effects, but eugenol provides faster and more consistent contact toxicity in termites. Previous research has shown that EOs containing phenolic compounds, such as eugenol, generally exhibit higher efficacy against social insects, such as termites [33,34]. This reinforces the present findings that eugenol-rich oils—particularly from clove and cinnamon—are stronger candidates for termite management than anethole-based oils. These results corroborate prior reports on the insecticidal potential of EOs [21,23,41] and extend their relevance to *Microcerotermes*, a genus that has been largely understudied. By linking chemical composition to bioassay results, this research provides solid evidence that eugenol-rich oils can be a reliable, eco-friendly alternative to synthetic termiticides.

Several reports have confirmed that the essential oil of anise (*Pimpinella anisum*) contains anethole as its principal component, usually ranging from 75% to 95% [42,43,44,45]. The variation in concentration depends largely on the plant material’s origin and the extraction methods used. These values are similar to star anise, which generally contains 85–90% anethole [46]. However, our study showed that anise oil had weaker termiticidal effects than star anise oil. This discrepancy suggests that the bio-efficacy of EOs cannot be determined solely by anethole content but may also depend on the presence of synergistic or antagonistic minor constituents, chemotypic variation, and environmental or extraction-related factors that influence oil quality. These factors could explain why star anise oil showed stronger biological activity despite having a composition similar to that of anise oil.

It is also noteworthy that several EOs—such as turmeric, phlai, sweet basil, peppermint, cajeput tree, kaffir lime, lemongrass, citronella grass, Siam cardamom, and eucalyptus—showed only low or very low mortality in *M. crassus* workers, particularly at lower concentrations. These results emphasize that not all botanicals with known insecticidal activity in other pests necessarily translate into strong termiticidal effects. For example, turmeric and phlai rhizome oils have been reported to kill stored-product insects and lice [35], but in this study, their effects on termites were limited. Similarly, oils rich in monoterpenes, have shown less consistent contact toxicity [21,23]. The low efficacy observed here underscores the species-specific nature of essential oil bioactivity and highlights that termite control requires compounds with rapid penetration and potent neurotoxic properties, such as eugenols. This further supports the conclusion that clove and cinnamon oils are the most promising candidates for termite management, while other oils may only serve as supplementary options.

Alongside contact toxicity measured under closed-system conditions, distinct patterns of repellency were also detected among the tested EOs. Clove and cinnamon oils, both rich in eugenols, showed the strongest and most consistent repellency against *M. crassus* workers, suggesting that eugenol not only acts as a toxicant but also interferes with termite orientation and foraging behavior. This dual mode of action has been reported previously in stored-product beetles and cockroaches, where eugenol-based oils were shown to disrupt olfactory-mediated host location [30,33]. Because the assays were conducted in a closed system, observed mortality may likely reflect combined contact and vapor-phase exposure rather than contact toxicity alone. The present study clearly demonstrated that clove and cinnamon oils exhibited significant repellency against *M. crassus* workers, particularly at higher concentrations. Previous studies have reported that EOs such as citronella grass, lemongrass, peppermint, and eucalyptus exhibit strong repellency against mosquitoes and other insect pests [47,48]. However, whether these effects apply to termites remains uncertain, because their physiology and behavior differ substantially. This indicates that repellency is highly species-specific, and compounds effective against Diptera or Coleoptera may not necessarily work against Isoptera. In addition, because repellency was measured repeatedly on the same worker groups, fine-scale temporal patterns should be interpreted with caution, as the time-series observations are not fully independent.

Several studies have shown that anethole has strong insect-repellent activity. For example, Alkan and Ertürk [49] reported high repellency against *Tribolium confusum*, while star anise and anise oils have been reported as moderately repellent to stored-product pests [31,32]. In contrast, the present study revealed that anethole exhibited low repellency against *M. crassus*. Interestingly, however, anethole showed distinct termiticidal activity, although it did not produce immediate contact mortality comparable to eugenol. This finding suggests that the combination of low repellency and delayed toxicity could be advantageous in developing toxic bait formulations for termite control, thereby providing a potential application in sustainable termite management strategies. Taken together, the combined analysis of repellency and toxicity highlights eugenol-rich oils as the most promising candidates, providing both lethal and deterrent effects. These results establish a foundation for the selection of effective EOs against termites, with future work focusing on their formulation into nanoemulsions to improve stability and field applicability. The contrasting activity patterns observed among the tested oils may indicate different potentials for future applications, although additional studies are required to verify these possibilities.

## 5. Conclusions

Eugenol-rich essential oils from clove and cinnamon exhibited strong contact toxicity and measurable repellency against *M. crassus* in closed-system laboratory assays, achieving complete mortality at 250 µL/L and over 80% repellency within one hour. In contrast, anethole from star anise oil exhibited slower but measurable termiticidal effects with low repellency, suggesting its potential use in slow-acting bait formulations. These findings provide preliminary evidence of termiticidal potential under laboratory conditions; however, soil-based, long-term, and field assays are required to assess their practical applicability. The results also support continued development of nanoemulsion delivery systems for sustainable termite management.

## Figures and Tables

**Figure 1 insects-16-01261-f001:**
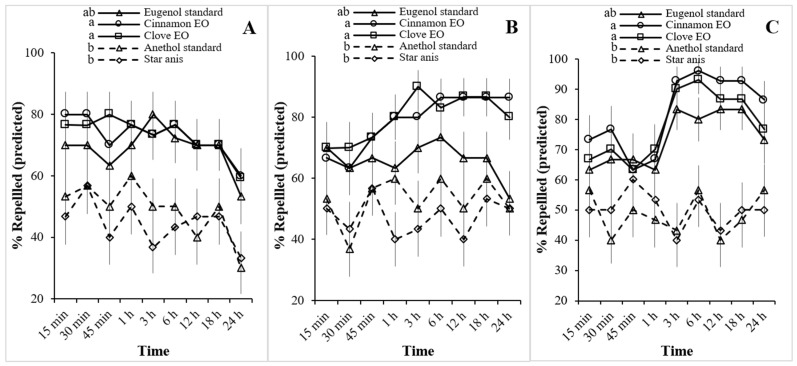
Predicted percentage of *Microcerotermes crassus* workers repelled by plant essential oils (EOs) and chemical standards, as analyzed using a binomial generalized linear model (GLM). Repellency was assessed by the contact bioassay at different exposure times and concentrations: (**A**) 50 µL/L, (**B**) 100 µL/L, and (**C**) 500 µL/L. Different letters above lines indicate significant differences among treatments within each concentration (Tukey’s HSD, *p* < 0.05).

**Table 1 insects-16-01261-t001:** Contact toxicity in a closed system of plant essential oils (EOs) against *Microcerotermes crassus* workers, evaluated 3 h after treatment.

Family	Common Name	Thai Name	Scientific Name	Part Used	Effecacy ^1^		
					Concentrations (µL/L)		
					10,000	1000	500
Apiaceae	Anise	Thein-Sat-Ta-Butr	*Pimpinella anisum*	Seeds	VH	H	-
Lauraceae	Cinnamon	Ob-Choei	*Cinnamomum zeylanicum*	Leaves	VH	VH	VH
Rutaceae	Kaffir lime	Ma-Krut	*Citrus hystrix*	Peel	VH	VL	-
Schisandraceae	Star anise	Chan-Paet-Klip	*Illicium verum*	Flower	VH	VH	M
Poaceae	Lemon grass	Ta-Khrai-Ban	*Cymbopogon citratus*	Stem and leaves	VH	L	-
	Citronella grass	Ta-Khrai-Hom	*Cymbopogon nardus*	Stem and leaves	VH	L	-
Lamiaceae	Peppermint	Sa-Ra-Nae	*Mentha* spp.	Leaves	VH	VL	-
	Sweet basil	Ho-Ra-Pha	*Ocimum basilicum*	Leaves	VH	VL	-
Myrtaceae	Cajeput tree	Sa-Met	*Melaleuca leucadendron*	Leaves	VH	VL	-
	Clove	Kan-Phlu	*Syzygium aromaticum*	Buds	VH	VH	VH
	Blue gum	Eucalyptus	*Eucalyptus camaldulensis*	Leaves	VH	VL	-
Zingiberaceae	Siam cardamom	Kra-Wan	*Amomum krervanh*	Roots	VH	VL	-
	Phlai	Phlai	*Zingiber cassumunar*	Roots	VH	VL	-
	Turmeric	Khamin-Chan	*Curcuma longa*	Roots	H	VL	-

^1^ Classification of mortality; VL: Very low < 10%, L: Low 10–25%, M: Medium 26–50%, High 51–75% and VH: Very high > 75%.

**Table 2 insects-16-01261-t002:** Major chemical constituents of clove, cinnamon, and star anise essential oils identified by GC-MS/MS analysis.

Chemical Components	%	Formula	Molecular Weight (g/mol)
*Clove essential oil*			
Eugenol	66.69	C_10_H_12_O_2_	164.20
Caryophyllene	17.34	C_15_H_24_	204.35
Humulene	1.13	C_15_H_24_	204.35
*α*-Bisabolene	6.22	C_15_H_24_	204.35
Cadinene	2.74	C_15_H_24_	204.35
Neoclovene oxide	1.20	C_15_H_24_O	220.35
Betulenol	1.39	C_30_H_50_O_2_	442.72
Other	3.39		
*Cinnamon essential oil*			
*α*-Pinene	2.44	C_10_H_16_	136.23
Carene	0.38	C_10_H_16_	136.23
*p*-Cymene	2.26	C_10_H_14_	134.22
Linalool	2.96	C_10_H_18_O	154.25
Cinnamaldehyde	1.75	C_9_H_8_O	132.16
Safrole	2.54	C_10_H_10_O_2_	162.19
Eugenol	54.99	C_10_H_12_O_2_	164.20
Caryophyllene	7.22	C_15_H_24_	204.35
Cinnamyl acetate	3.88	C_11_H_12_O_2_	176.21
Humulene	2.06	C_15_H_24_	204.35
Caryophyllene oxide	2.00	C_15_H_24_O	220.35
Benzyl Benzoate	5.66	C_14_H_12_O_2_	212.24
Other	11.86		
*Star anise essential oil*			
Linalool	1.39	C_10_H_18_O	154.25
Anisic aldehyde	1.81	C_8_H_8_O_2_	136.15
Anethole	90.78	C_10_H_12_O	148.20
α-Copaene	0.81	C_15_H_24_	204.35
1-(3-Methyl-2-butenoxy)-4-(1-propenyl)benzene	1.23	C_14_H_18_O	202.29
Other	3.98		

**Table 3 insects-16-01261-t003:** Mortality percentages and toxicity values (LC_50_ and LC_90_) of plant essential oils (EOs) against *Microcerotermes crassus* workers. Mortality was evaluated at 3 h intervals over a 24 h exposure period using contact method at different EO concentrations.

Treatments/ After Treated (h)	% Mortality ^1^ (Means)					Regression Equation ^2^	SE	χ^2^	Toxicity Values ^3^ (µL/L)	
	Concentrations (µL/L)									
	0	100	250	500	750				LC_50_	LC_90_
*Clove EO*										
3 h	0.0 d	13.3 c	66.7 b	100.0 a	100.0 a	Y = 0.011 − 2.358x	0.229	1.606	208.6	322.0
6 h	0.0 c	30.0 b	96.7 a	100.0 a	100.0 a	Y = 0.017 − 2.313x	0.229	1.674	135.5	210.5
12 h	0.0 c	60.0 b	100.0 a	100.0 a	100.0 a	Y = 0.039 − 3.630x	2.240	0.017	93.6	126.7
18 h	0.0 c	70.0 b	100.0 a	100.0 a	100.0 a	Y = 0.042 − 3.634x	2.256	0.018	87.6	118.5
24 h	0.0 b	93.3 a	100.0 a	100.0 a	100.0 a	Y = 0.050 − 3.561x	1.996	0.023	70.6	96.0
*Cinnamon EO*										
3 h	0.0 c	0.0 c	20.0 b	86.7 a	100.0 a	Y = 0.009 − 3.109x	0.263	2.323	362.6	512.1
6 h	0.0 c	0.0 c	43.3 b	100.0 a	100.0 a	Y = 0.023 − 5.922x	3.646	0.024	256.8	312.4
12 h	0.0 c	3.3 c	60.0 b	100.0 a	100.0 a	Y = 0.014 − 3.272x	0.393	0.070	231.8	322.6
18 h	0.0 d	13.3 c	76.7 b	100.0 a	100.0 a	Y = 0.013 − 2.486x	0.243	0.934	191.9	290.8
24 h	0.0 c	20.0 b	100.0 a	100.0 a	100.0 a	Y = 0.028 − 3.700x	0.960	0.073	130.5	175.7
*Star anise EO*										
3 h	0.0 c	0.0 c	0.0 c	46.7 b	100.0 a	Y = 0.014 − 7.039x	2.398	0.069	507.0	599.3
6 h	0.0 d	6.3 cd	13.3 c	50.0 b	100.0 a	Y = 0.006 − 2.498x	0.182	16.055	450.3	681.3
12 h	0.0 d	6.7 d	23.3 c	60.0 b	100.0 a	Y = 0.005 − 2.226x	0.162	10.403	408.1	643.1
18 h	0.0 d	13.3 d	36.7 c	63.3 b	100.0 a	Y = 0.005 − 1.779x	0.134	17.995	365.4	628.7
24 h	0.0 e	16.7 d	43.3 c	70.0 b	100.0 a	Y = 0.005 − 1.635x	0.128	18.050	332.8	593.7
*Eugenol standard*										
3 h	0.0 c	0.0 c	20.0 b	100.0 a	100.0 a	Y = 0.019 − 5.524x	2.481	0.019	294.9	362.9
6 h	0.0 c	0.0 c	46.7 b	100.0 a	100.0 a	Y = 0.023 − 5.799x	3.039	0.047	252.8	308.6
12 h	0.0 d	13.3 c	70.0 b	100.0 a	100.0 a	Y = 0.012 − 2.394x	0.234	1.347	202.9	311.6
18 h	0.0 d	20.0 c	86.7 b	100.0 a	100.0 a	Y = 0.014 − 2.329x	0.221	1.517	167.0	258.9
24 h	0.0 c	43.3 b	100.0 a	100.0 a	100.0 a	Y = 0.032 − 3.338x	1.392	0.082	104.5	144.6
*Anethole standard*										
3 h	0.0 d	0.0 d	16.7 c	66.7 b	100.0 a	Y = 0.007 − 2.932x	0.239	5.873	418.8	601.8
6 h	0.0 c	0.0 c	26.7 b	100.0 a	100.0 a	Y = 0.020 − 5.520x	3.039	0.024	281.2	346.5
12 h	0.0 d	16.7 c	36.7 b	100.0 a	100.0 a	Y = 0.008 − 2.251x	0.179	25.814	276.9	434.5
18 h	0.0 d	26.7 c	43.3 b	100.0 a	100.0 a	Y = 0.008 − 1.813x	0.154	21.926	234.3	399.8
24 h	0.0 d	36.7 c	53.3 b	100.0 a	100.0 a	Y = 0.008 − 1.796x	0.160	14.359	216.6	371.1

^1^ Data were determined based on n = 10 adults of working termites/three replications, means within the same row followed by the same letter are not significantly different (*p* < 0.05) according to DMRT. ^2^ Probit (Y) = Intercept + Slope · (concentration: x). ^3^ Lethal concentrations of plant essential oils (EOs) needed to kill 50% and 90% of the termites (LC_50_ and LC_90_, respectively) at different hours after treatment. Only point estimates are reported because 95% CIs were not available.

**Table 4 insects-16-01261-t004:** Mortality percentages and lethal time values (LT_50_ and LT_90_) of plant essential oils (EOs) against *Microcerotermes crassus* workers at different concentrations, as determined by the contact method over exposure periods.

Treatments/ Concentrations (µL/L or mg/L)		% Mortality ^1^ (Means)					Regression Equation ^2^	SE	χ^2^	Toxicity Values ^3^ (h)	
		After Various Exposure Times (h)								LT_50_	LT_90_
		3	6	12	18	24					
*Clove EO*											
	100	13.3 d	30.0 cd	60.0 bc	70.0 ab	93.3 a	Y = 0.012 − 1.274x	0.126	6.621	11.42	22.91
	250	66.7 b	96.7 a	100.0 a	100.0 a	100.0 a	Y = 0.469 − 0.975x	0.355	<0.001	2.08	4.81
*Cinnamon EO*											
	100	0.0 c	0.0 c	3.3 bc	13.3 ab	20.0 a	Y = 0.096 − 3.038x	0.333	3.259	31.65	45.00
	250	20.0 d	43.3 c	60.0 bc	76.7 b	100.0 a	Y = 0.116 − 1.083x	0.124	12.877	9.31	20.32
*Star anise EO*											
	100	0.0	6.3	6.7	13.3	16.7	Y = 0.050 − 2.097x	0.200	5.21	41.91	67.51
	250	0.0	13.3	23.3	36.7	43.3	Y = 0.072 − 1.750x	0.152	11.983	24.283	42.062
*Eugenol standard*											
	100	0.0 c	0.0 c	13.3 b	20.0 b	43.3 a	Y = 0.109 − 2.752x	0.245	7.752	25.19	36.92
	250	20.0 e	46.7 d	70.0 c	86.7 b	100.0 a	Y = 0.134 − 1.083x	0.128	7.429	8.07	17.63
*Anethole standard*											
	100	0.0	0.0	16.7	26.7	36.7	Y = 0.092 − 2.441x	0.213	13.143	25.91	39.52
	250	16.7	26.7	36.7	43.3	53.3	Y = 0.045 − 0.968x	0.119	1.638	21.41	49.76
*Insecticides (Recommended dose)*											
Fipronil	250	43.3 b	90.0 a	100.0 a	100.0 a	100.0 a	Y = 0.484 − 1.620x	0.304	0.001	3.35	6.00
Cypermethrin	2000	100.0 a	100.0 a	100.0 a	100.0 a	100.0 a	-			-	-

^1^ Data were determined based on n = 10 adults of working termites/three replications, means within the same row followed by the same letter are not significantly different (*p* < 0.05) according to DMRT. ^2^ Probit (Y) = Intercept + Slope · (concentration: x). ^3^ Lethal times of plant essential oils (EOs) needed to kill 50% and 90% of the termites (LT_50_ and LT_90_, respectively) at different hours after treatment. Only point estimates are reported because 95% CIs were not available.

## Data Availability

The original contributions presented in this study are included in the article. Further inquiries can be directed to the corresponding author.

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
