# Peer review of "In Vitro Termiticidal Activity of Medicinal Plant Essential Oils Against Microcerotermes crassus"

_insects, 2025, doi:10.3390/insects16121261_

Round 1
Reviewer 1 Report
Comments and Suggestions for Authors
This paper deals with essential oils from various plant sources, analyzes their components, and examines their insecticidal and repellent effects using Macrotermes sp. This study investigates insecticidal activity by confining termites in containers and forcing them to contact with essential oils all the time. If sufficient quantities and qualities the oils would keep at the source, the oils would show the repellency. Nevertheless, essential oils readily volatilize or undergo chemical changes in their constituents, diminishing their repellent effect.
The selected biological assay method demonstrated the intended termite resistance, and the results obtained are considered reliable. However, apart from the termite species being Macrotermes sp., the novelty seems to be limited.
Please provide responses to the following points for comment.
(1) How was the termite's lethal state determined? (For example, it was determined to be lethal even if, after one minute of observation, it showed no movement, including of its antennae or legs, or if it was lying on its back but its antennae were still moving, etc.). Please describe in the text.
(2)The argument that the essential oils are safe simply because they are natural substances is incorrect. Some reports are there on the fish toxicity and mammalian toxicity of eugenol. Please revise the introduction to address this point as well.
(3)Macrotermes sp. identification was not performed. Even among termites of the same genus, the effectiveness of synthetic pesticides varies. Why was species identification not performed ?
Author Response
Comments 1: How was the termite's lethal state determined? (For example, it was determined to be lethal even if, after one minute of observation, it showed no movement, including its antennae or legs, or if it was lying on its back but its antennae were still moving, etc.). Please describe it in the text.
Response: We appreciate this suggestion. The Methods section now specifies that workers were classified as dead when they could not stand or walk after gentle probing, even if slight antennal or leg movements remained (line 158-161).
Comments 2: The argument that essential oils are safe simply because they are natural substances is incorrect. Some reports are here on the fish toxicity and mammalian toxicity of eugenol. Please revise the introduction to address this point as well.
Response: The Introduction has been revised to clarify that EO safety is dose-dependent, and supporting evidence of eugenol toxicity in fish and mammals has been added (lines 83–86; Tao et al., 2023; Pramod et al., 2024). The corresponding references have also been added to the reference list (Line 506-508 and 534-536).
Comments 3: Macrotermes sp. identification was not performed. Even among termites of the same genus, the effectiveness of synthetic pesticides varies. Why was species identification not performed?
Response: Thank you for the comment. Species identification was not possible at the initial submission because only workers were available, and their morphology was insufficient for reliable identification. Following the reviewer’s suggestion, additional specimens were examined, and the termite was confirmed as Microcerotermes crassus. This information has been added to the revised manuscript.

Reviewer 2 Report
Comments and Suggestions for Authors
This manuscript adds to the growing body of literature evaluating essential oils for their utility in pest management systems. The introduction focuses on two other genera of termites (not the study organism); this needs to be corrected or explained before publication. Also, the methodology does not easily distinguish between contact toxicity, fumigant toxicity, and repellency; this needs to be corrected or explained before publication. Finally, the conclusions about utility of these EOs as termiticides are not warranted from the results. Please retract or reframe as conjecture needing further research.

Please secure a review of the entire text from a native English speaker, with special attention to grammar, subject-verb agreement, spelling, and style.
Author Response
Comments 1: The introduction focuses on two other genera of termites (not the study organism); this needs to be corrected or explained before publication.
Response: Thank you for this valuable comment. The Introduction has been thoroughly revised to focus specifically on Microcerotermes crassus, the study organism. Previous content emphasizing other termite genera (e.g., Coptotermes and Macrotermes) has been removed or replaced with ecological and economic information directly relevant to Microcerotermes, supported by national survey data (Lertlumnaphakul et al., 2022). These revisions ensure that the background context aligns appropriately with the species investigated in this study (lines 51–59, 64-66).
Comments 2: The methodology does not easily distinguish between contact toxicity, fumigant toxicity, and repellency; this needs to be corrected or explained before publication.
Response: Thank you for this important comment. The contact-toxicity assays in our study were conducted as contact–residue tests in sealed Petri dishes. We acknowledge that sealed-dish conditions may allow a limited contribution of vapor-phase exposure, and therefore, the revised Methods now describe the assay conditions more transparently by explicitly indicating that the tests were performed as contact assays under closed-system conditions (Lines 29, 103-104, 150, 163, 176, 225 and 301). For the repellency assays, we have also clarified that they were conducted under the same closed-system setup, and that repellency was determined exclusively from the spatial distribution of live termites on the treated and untreated halves of the filter paper. Mortality was not used to infer repellent responses (Lines 184–186). These revisions enhance the clarity of our methodology and ensure an accurate distinction between contact toxicity, vapor-related effects, and behavioral repellency.
Comments 3: The conclusions about utility of these EOs as termiticides are not warranted from the results. Please retract or reframe as conjecture needing further research.
Response: Thank you for this comment. The Conclusions section has been revised to avoid overinterpretation. Statements implying direct termiticide utility have been softened, and the text now emphasizes that the findings represent preliminary laboratory evidence requiring further soil-based and field studies (lines 393–395).
Comments 4: Lines 38-40: In general, there are numerous grammatical and style-related mistakes and inconsistencies. I suggest that the authors consult with a native English speaker to review and revise entire manuscript.
Response: We revised the Abstract to avoid overinterpretation and now use cautious, evidence-based language that reflects preliminary laboratory trends only (lines 38–41).
Comments 5: Lines 50-51: check ecological facts: Coptotermes does not build mounds, and Maxrotermes is not considered a subterranean termite genus.
Response: We appreciate the reviewer’s correction. The previous text inaccurately described Coptotermes and Macrotermes ecology. This section has now been fully revised to remove the incorrect statements and to align the Introduction with the biology of Microcerotermes crassus, the actual study organism. Updated content is based strictly on reliable sources (e.g., Lertlumnaphakul et al., 2022), and the erroneous descriptions of mound-building and subterranean classification have been deleted (lines 51–59).
Comments 6: Lines 93: this is the first mention of Microcerotermes. This species should be established as economically important.
Response: We appreciate the reviewer’s comment. The Introduction has been revised to clearly establish Microcerotermes—specifically M. crassus—as an economically important structural and agricultural pest in Thailand. Supporting ecological and economic evidence from Lertlumnaphakul et al. (2022) has been added (lines 51–59) to ensure that the significance of this species is fully described before its first mention.
Comments 7: Lines 110: is eugenol not an EO? Please explain the difference or correct
Response: Thank you for this valuable comment. In the revised manuscript, we clarified that the stock solutions were prepared from aqueous dispersions of each essential oil as well as from each purified chemical standard (eugenol and anethole). This revision provides a clearer distinction between whole essential oils and their individual component standards and resolves the ambiguity present in the previous version. (Lines 117–118)
Comments 8: Lines 146-147: if arena dishes were "closed", then we do not know if mortality was due to contact or fumigation (volatile toxicity)
Response: Thank you for highlighting this methodological limitation. We acknowledge that conducting assays in closed Petri dishes does not allow the separation of contact toxicity from potential fumigant (vapor-phase) effects. In the revised manuscript, we have clarified this limitation in the Methods by explicitly describing the assays as closed-system tests (lines 150, 163), and we have added a statement in the Discussion noting that observed mortality may reflect combined contact and vapor-phase exposure rather than contact toxicity alone (lines 363-364).
Comments 9: Lines 367-368: the leap to bait formulation is unfounded. This statement is conjecture and should be omitted or clearly identified as such.
Response: We thank the reviewer for highlighting this concern. The previous sentence referring to potential use in bait formulations has been removed to avoid overinterpretation. In its place, we added a more conservative statement acknowledging that the contrasting activity profiles of the oils may imply different application potentials, while explicitly noting that further studies are required to verify these possibilities (lines 393–395).
Reviewer 3 Report
Comments and Suggestions for Authors
- Some abbreviations should be fully written when first mentioned.
- The term "ppm" is no longer commonly used; it is recommended to use "mg/L".
- Please specify the size of worker ants selected in the methods section, as this is crucial for experimental results.
- In the 3.3 section, the statement "These findings indicate that clove and cinnamon oils are far more effective repellents than star anise oil and anethole" should be accompanied by a significance analysis of the results from Figure 1.
Author Response
Comments 1: Some abbreviations should be fully written when first mentioned.
Response: We thank the reviewer for this helpful suggestion. All abbreviations have now been fully written out at their first occurrence throughout the manuscript to ensure clarity and compliance with journal style requirements (revised throughout).
Comments 2: The term "ppm" is no longer commonly used; it is recommended to use "mg/L".
Response: We appreciate the reviewer’s suggestion. In the revised manuscript, all concentration units previously expressed in ‘ppm’ have been replaced with SI-compliant units. Essential oil concentrations are now reported in µL/L (v/v), and synthetic insecticide are reported in mg/L, in accordance with current conventions.
Comments 3: Please specify the size of worker ants selected in the methods section, as this is crucial for experimental results.
Response: The Methods section has been revised to specify the size of worker Microcerotermes crassus used in the experiments. Termite workers measuring approximately 4.7 ± 0.4 mm in body length were selected (line 132-133).
Comments 4: In the 3.3 section, the statement "These findings indicate that clove and cinnamon oils are far more effective repellents than star anise oil and anethole" should be accompanied by a significance analysis of the results from Figure1.
Response: hank you for this valuable suggestion. We have added a post hoc significance analysis using Tukey’s HSD applied to estimated marginal means (EMMs) from the GLM model. The Data Analysis section has been updated accordingly (lines 203-205), and statistically homogeneous groups are now shown as letter labels above each treatment line in Figure 1 and lines 298-299. The Results section has also been revised to reflect these analyses (lines 281–285).